# Experimental and Finite Element Simulation of Polyolefin Elastomer Foams Using Real 3D Structures: Effect of Foaming Agent Content

**DOI:** 10.3390/polym14214692

**Published:** 2022-11-03

**Authors:** Ehsan Rostami-Tapeh-Esmaeil, Amirhosein Heydari, Ali Vahidifar, Elnaz Esmizadeh, Denis Rodrigue

**Affiliations:** 1Department of Chemical Engineering, Université Laval, Quebec, QC G1V 0A6, Canada; 2Department of Polymer Science and Engineering, University of Bonab, Bonab 5551761167, Iran; 3Construction Research Center, National Research Council Canada, Ottawa, ON K1A 0R6, Canada

**Keywords:** polyolefin elastomer, foams, morphology, mechanical properties, finite element method

## Abstract

In this study, polyolefin elastomer (POE) foams were prepared without any curing agent using a single-step foaming technique. The effect of azodicarbonamide (ADC) content as a chemical foaming agent on the foams’ morphology and mechanical properties was studied using scanning electron microscopy (SEM), mechanical properties (tension and compression) and hardness. The results showed that increasing the ADC content from 2 to 3, 4 and 5 phr (parts per hundred rubber) decreased the foam density from 0.75 to 0.71, 0.65 and 0.61 g/cm^3^, respectively. The morphological analysis revealed that increasing the ADC content from 2 to 4 phr produced smaller cell sizes from 153 to 109 µm (29% lower), but a higher cell density from 103 to 591 cells/mm^3^ (470% higher). However, using 5 phr of ADC led to a larger cell size (148 µm) and lower cell density (483 cells/mm^3^) due to cell coalescence. The tensile modulus, strength at break, elongation and hardness properties continuously decreased by 28%, 21%, 16% and 14%, respectively, with increasing ADC content (2 to 5 phr). On the other hand, the compressive properties, including elastic modulus and compressive strength, increased by 20% and 64%, respectively, with increasing ADC content (2 to 5 phr). The tensile and compression tests revealed that the former is more dependent on foam density (foaming ratio), while the latter is mainly controlled by the cellular structure (cell size, cell density and internal gas pressure). In addition, 2D SEM images were used to simulate the foams’ real 3D structure, which was used in finite element methods (FEM) to simulate the stress–strain behavior of the samples at two levels: micro-scale and macro-scale. Finally, the FEM results were compared to the experimental data. Based on the information obtained, a good agreement between the macro-scale stress–strain behavior generated by the FEM simulations and experimental data was obtained. While the FEM results showed that the sample with 3 phr of ADC had the lowest micro-scale stress, the sample with 5 phr had the highest micro-scale stress due to smaller and larger cell sizes, respectively.

## 1. Introduction

Thermoplastic elastomers (TPEs) are copolymers combining the elasticity of rubbers with the recyclability and processing advantages of thermoplastics [1,2]. TPEs can easily flow when heated, especially under shear, while regaining a new structure and stability after cooling [3,4]. Unlike the chemical crosslinks in thermoset rubbers, TPEs involve physical crosslinking, which can be reversed by heating, making it possible to re-use/recycle all the production waste and the parts after their end of life [5]. TPEs can be classified into styrene block copolymers (TPE-S or TPS), thermoplastic vulcanizates (TPE-V or TPV), thermoplastic polyesters (TPE-E or TPC), thermoplastic polyurethanes (TPE-U or TPU), thermoplastic polyamides (TPE-A or TPA) and thermoplastic polyolefins (TPE-O, TPO or POE) [6,7,8,9]. POEs, as a more recent class of TPEs, were developed from recent improvements in metallocene polymerization catalysts based on polyolefins (polyethylene, polypropylene or their copolymers) [10,11]. The majority of commercially available POEs are ethylene-butene or ethylene-octene copolymers. They can be easily processed by injection molding, extrusion or blow molding. Because of their high mechanical characteristics (resilience and toughness) and easy processing, these materials are now widely used in several applications such as sheets, automotive components, durable products, engineering plastics modifiers, as well as wires and cables [12]. Today, low cost, weight reduction, raw material savings, thermal insulation, chemical resistance, noise reduction, sealing applications and any applications where energy absorption or flexibility is required are of interest, especially using polymer foams [13,14,15]. POE foams can be found in a wide range of applications including seals, insulation, construction, cushions, damping and impact absorbers [16].

Rubber foams can be expanded in either one or two steps. In the single-step foaming process, the compound is placed in a mold and kept at high temperature and high pressure to simultaneously cure and foam the sample. Then, the mold is cooled to ambient temperature to stabilize the foam morphology [17]. In the two-step foaming process, the compound is first placed in a press and preheated at a temperature below the foaming agents’ decomposition to pre-cure and pre-form the compound without foaming agent decomposition, while in the second step the sample is placed in a hot air oven for foaming and to complete the curing [18,19]. In general, the single-step approach generates foams with more precise dimensions, making it easier to control the cell density/size and mechanical properties [20]. As expected, processing conditions such as temperature, time and pressure, in addition to formulation parameters (type and concentration of filler and foaming agent) play critical roles in the final foam properties [21]. This is why it is important to control the formulation and processing step to optimize the final foam structure, which controls the physical and mechanical behavior. In our previous work, we achieved functionally graded POE foams by controlling the processing temperature via the single-step foaming process [22]. We were able to obtain a range of thermal insulation properties (0.125 to 0.180 W/m.K) by controlling the cellular structure during the foaming process, which was effective for thermal insulation applications.

Chemical foaming agents are generally solids (powders) generating gases once they reach their thermal decomposition temperature [23]. Azodicarbonamide (ADC), with a decomposition temperature of around 210 °C (pure ADC), is one of the most famous and versatile chemical foaming agents [24]. The main reason is due to the possibility of controlling its decomposition temperature using activators (zinc oxide (ZnO) and zinc stearate (ZnSt)) combined with high gas yield efficiency (231 cm^3^ of gas per gram) [25,26]. 

In the presence of NaCl (as a pore-forming agent), Peng et al. improved the sound absorption efficiency of silicon rubber (SR) foams for the middle frequency range (1000–2000 Hz). Their findings were related to the improved resonance matching induced by the open-cell foam structure [27]. Natural rubber (NR) foams enhanced by a physical hybrid of nanographene (GNS) and carbon nanotubes (CNT) were produced by Shojaei and co-workers [28]. According to their results, the simultaneous increase in GNS and CNT content from 0 to 2 phr reduced the cell size from 800 to 80 µm, enhanced the compression behavior and raised the stress at 50% strain from 150 kPa to 440 kPa.

Hassan and co-workers prepared ethylene propylene diene monomer (EPDM) foams loaded with various ADC contents (0, 5, 10 and 15 phr) using two different crosslinking systems (peroxide and sulfur) [29]. According to their results, the sample with 15 phr of ADC had higher compressive characteristics, notably for peroxide-cured EPDM, while the swelling degree was reduced for EPDM that was vulcanized by sulfur. In addition, tensile tests revealed higher tensile and Young’s modulus for unfoamed samples using both curing systems. In a similar work on natural rubber (NR) foam, Charoeythornkhajhornchai et al. reported that increasing the foaming agent content decreased the foam density and mechanical characteristics (modulus and hardness) [30]. Mao et al. studied the physical and mechanical properties of POE foams, which were found to be mainly controlled by their density [31]. Hence, increasing the ADC concentration from 0 to 13 phr in the presence of titanium dioxide (TiO_2_) as a filler and dicumyl peroxide (DCP) as a curing agent resulted in a lower foam density from 0.896 to 0.057 g/cm^3^, leading to lower physical and mechanical properties. The effect of ADC concentration (0, 2, 4, 6, 8 and 10 phr) on the microstructure of EPDM rubber foams was also studied by Wang and co-workers [32,33]. They also reported that increasing the ADC content increased both the cell density and cell size. To further study the effect of foaming agent content on the density and mechanical properties of nitrile-butadiene rubber (NBR) and EPDM foams, Lawindy et al. used ADC contents up to 15% [34]. The density of both NBR and EPDM rubber foams reduced as the foaming agent concentration increased. The effect of ADC content (up to 12.5 phr) on NBR foams was also studied by Mahmoud et al. [35]. The foams’ compressive stress–strain behavior revealed a very good agreement with the Gibson–Ashby model by decomposition in two parts: linear (elastic) and non-linear (curved) regions [36].

Even though several studies on the manufacturing and characterization of rubber foams have been published, there is limited information on how to model or simulate their structure–property relationship, which is useful for design and optimization [37]. This is especially the case for numerical simulations used to investigate and predict the mechanical behavior of these materials under various conditions, and no information on the effect of foaming agent content on the properties of POE foams can be found. The main reason for this is the difficulty associated with the complex morphology and the high number of parameters to account for. The precise foam structure must be simulated to conduct representative FEM calculations. Previous studies reported that two methods are available for the simulation of foamed morphologies: real and ideal models. Ideal models are based on a uniform distribution of four distinct structures: (a) the technique of random polyhedral Voronoi or the Weaire–Phelan structure (polystyrene (PS) [38], metal [39], closed cell foams [40], polyethylene (PE) and PS foams [41]); (b) randomly packed spheres [42]; (c) polyhedral geometry using experimental data (polyurethane (PU) [43], EPDM [44] and PU foams [45]); and (d) the distribution of spheres using an experimentally determined morphology (carbon foam [46]). However, very few works used the real morphology of non-rubber materials such as expanded PS [47], aluminum [48], auxetic material based on PU [49], pitch-based carbon [50], and metallic foams [51]. Furthermore, only a few studies have integrated FEM with other numerical theories and types of elements, such as the bar and shell [45] or solid tetrahedral elements [52]. In order to predict the behavior of any foam, the effect of relative density, cell size distribution and cell configuration was not taken into account. On the other hand, the differential quadrature and Bezier numerical techniques, which have a relatively recent origin, demonstrated more stability and accuracy than the other numerical methods for the initial- and/or boundary-value problems of physical and engineering sciences [53,54].

Numerous reports have been published on the simulation of polymeric foams, but very few of them have conducted a FEM analysis on thermoplastic elastomer foams, especially POE foams. In addition, most of the works on POE foams included curing agents to improve the final mechanical characteristics and control the foaming process (nucleation and growth steps). In this study, however, POE foams with various ADC contents (2 to 5 phr) were prepared in the absence of a curing agent via a single-step foaming process. The morphological, physical and mechanical properties of POE foams were determined by SEM, tension, compression and hardness. Then, using 2D SEM images, the true 3D foam structure was simulated using a layer-by-layer analysis and the foams’ tensile behavior and investigated using FEM. Finally, the experimental data were used to validate the FEM results at the macro-scale level and the results were compared to different hyper-elastic models. The FEM findings are also given and analyzed at the micro-scale level to determine the effect of the cell structure on the localized stress–strain relationships.

## 2. Materials and Methods

### 2.1. Materials

Poly(ethylene-*co*-octene) (PEO Engage 8150) containing 24 wt.% of octene with a melt index of 0.50 g/10 min (190 °C/2.16 kg, ASTM D1238), Mooney viscosity of 33 (ML 1 + 4, 121 °C, ASTM D1646), and density of 0.870 g/cm^3^ was supplied by the Dow Chemical Company (Midland, MI, USA). Azodicarbonamide (ADC) (AZ-760 A), as an exothermic chemical foaming agent with a decomposition temperature range between 185 and 210 °C, was purchased from Chempoint (Bellevue, WA, USA).

### 2.2. Preparation of POE-ADC Compounds

POE compounds with different ADC concentrations (0, 2, 3, 4 and 5 phr) were mixed via a co-rotating twin-screw extruder (ZSE-Leistritz 27/40D). All the processing conditions, such as feeding rate (2 kg/h), screw speed (12 rpm) and average barrel temperature (110 °C), were constant. The extruded compounds passed in an ice bath to cool down before being cut into pellets. Finally, to remove residual stresses, the samples were kept at ambient temperature for at least 24 h. The samples’ formulations and codes are presented in Table 1.

### 2.3. Foams Preparation and Characterization

POE foams were prepared via the single-step foaming process. Firstly, 10 g of each compound was placed in a mold (8 cm in length, 5 cm in width and 0.35 cm in thickness). Then, the mold was placed in an electrically heated hot press for 12 min at 205 °C under 8.5 bar to reach complete ADC decomposition and gas diffusion (uniform). Before opening for expansion, the mold was cooled down under pressure to ambient temperature to stabilize the cell structure.

Density was obtained via a digital caliper (Mastercraft, Ottawa, ON, Canada) to determine the volume, while the weight was measured by a balance (MX-50, A&D, Ann Arbor, MI, USA). The average of three determinations was used to report each result. The foam densities obtained were 0.75, 0.71, 0.65 and 0.61 g/cm^3^ for 2, 3, 4 and 5 phr of ADC, respectively. The foaming ratio was calculated according to [55]:(1)$$\mathrm{Foaming}\;\mathrm{ratio}\;\left(\%\right)=\left(1-\frac{Density\;of\;foamed\;sample}{Density\;of\;unfoamed\;sample}\right)\times 100$$

To examine the foam morphology and cell structure, each sample was cut into several layers and SEM images (FEI Inspect F50, Hillsboro, OR, USA) were taken at different magnification levels. The BELView software was used to perform quantitative determination of the foam structure. For a quantitative analysis of the resulting morphology, several parameters, such as the number average cell size (*D_n_*), weight average cell size (*D_w_*), polydispersity index (*PDI*) and cell density (*ρ_cell_*) were used, as shown in Table 2.

## 3. Results and Discussion

### 3.1. Morphological and Physical Characterization

The SEM images and cell size distribution of the foamed POE with different ADC contents are presented in Figure 1a,b, respectively. The quantitative characterizations of these SEM images and their physical properties are depicted in Figure 2. According to the results, increasing the ADC content from 2 to 4 phr led to a lower cell size from 153.1 to 109.4 µm (29%) and higher cell density from 103 to 591 cells/mm^3^ (470%). Moreover, the cell size uniformity was changed, as the cell size range for PA2 was (94–215) μm, while it was (58–171) μm for PA4 (Figure 1b). This indicates that in contrast to samples with a low ADC concentration (2 phr), increasing the foaming agent content (4 phr) resulted in narrower cell size distributions and lower PDI, and thus, more uniform/homogenous samples. In this regard, both nucleation and growth steps were affected by the foaming agent concentration. A higher ADC content led to the generation of more nuclei, thus, increasing the cell density by 470%. On the other hand, in the single-step foaming process, as used in this study, the mold was closed during the foaming and stabilization steps. Therefore, the foams cannot exceed the mold volume. As a result, the cell growth or cell size of PA4 was more limited by nucleation and mold size, leading to a higher cell density and lower cell size (higher gas content and more restrictions). On the contrary, for the sample with 5 phr of ADC, the cell size increased to 148.1 µm (35%), while the cell size range increased to (66–223) µm and the cell density decreased to 483 cells/mm^3^ (18%). Since the ADC content increased from 4 to 5 phr, a higher volume of gas is available to create higher internal cell pressure, increasing the possibility of cell coalescence and collapse. There is also a higher plasticization effect of the dissolved gas molecules decreasing the matrix melt elasticity and viscosity, which is no longer able to prevent the cell walls from rupturing during processing [59]. In addition, Figure 1 clearly shows that PA5 has the highest PDI with an elliptical/elongated cell, while decreasing the ADC content made the foams more uniform (lower PDI) with more spherical cells. Both phenomena of cell structure and PDI are related to cell coalescence and cell wall rupture. In other words, at a lower ADC content, the gas volume and gas pressure generated by the ADC decomposition were not enough to significantly change the cell shape and their breakup. This is not the case for PA5 where deformed cells are observed [60].

Figure 2 also illustrates the foam density and foaming ratio. The density of the unfoamed sample (PA0) is 0.874 g/cm^3^, but when the ADC concentration increased from 2 to 5 phr the foam density decreased from 0.748 to 0.612 g/cm^3^. This trend is related to the higher amount of gas generated with an increasing ADC content [61]. In other words, the higher gas volume from ADC resulted in a higher foaming ratio from 14.4 to 30.0% for 2 to 5 phr, respectively. The schematic representation of the relation between ADC content and cell density/cell size is presented in Figure 3.

### 3.2. Mechanical Properties

Figure 4 presents the mechanical properties of the POE foams. All the tensile parameters (modulus, strength and the elongation at break) of the foams (PA2 to PA5) are lower than the unfoamed (PA0) sample (Figure 4a). The addition of 2 phr of ADC into the matrix decreased the modulus, strength and elongation at break by 18%, 53% and 30%, respectively. This is mainly associated with the presence of a gas phase (bubbles/voids) inside the matrix [62]. However, other morphological and structural parameters, such as *D_n_*, *PDI* and *ρ_cell_*, can also control the mechanical properties [63,64]. According to Figure 4a, the tensile properties of the foams have the same trend, as they decreased with increasing the ADC content. The same trend is observed for the modulus and strength. For example, the modulus decreased from 0.91 to 0.66 MPa by increasing the ADC concentration from 2 to 5 phr. Mao et al. manufactured POE foams by using 5 phr of ADC (at 170 °C) with a cell size of 50.5 µm and a tensile strength of 2.02 MPa [31]. In our case, the tensile strength decreased by 21% (from 3.02 to 2.38 MPa) by the incorporation of a higher ADC content into the matrix (from 2 to 5 phr). This is attributed to the lower sample density (higher foaming ratio), as less material is available to sustain the applied stresses.

In contrast to tensile properties, the compressive stress–strain curves show a different trend with increasing the foaming agent content. According to the results of Figure 4b, the compressive characteristics (elastic modulus and compressive strength) increased with increasing the ADC content. This is associated to the higher ADC content resulting in more gases generated inside the matrix. The presence of more ADC not only changed the foam morphology (lower cell size and higher cell density, as reported in Figure 1 and Figure 2), but also led to higher gas volume/pressure inside the cells, improving their resistance against compressive forces because each cell acts as an inflated balloon. Hence, increasing the ADC from 0 to 5 phr improved the elastic modulus from 0.48 to 0.65 MPa, respectively. In addition, the compressive strength (at 7% compression) elevated from 0.013 to 0.023 MPa with enhancing the ADC concentration from 0 to 5 phr, respectively.

The hardness (Shore A) of the foams also decreased from 70.2 to 60.2 with a higher ADC content from 2 to 5 phr (Figure 4c). As the foaming ratio increased (with increasing ADC content), the rigidity and resistance of the foams against the needle penetration was reduced, leading to lower hardness.

## 4. Modeling and Simulation

### 4.1. Dimensional Geometry Modeling and Material Definition

Modeling of the 3D morphology or real foam structure is a crucial step in simulating the real foam behavior. The outcome accuracy is largely determined by the accuracy and precision with which the real 3D foam structure is represented. The SEM images were used to create 2D models at various levels. The real cells displayed in light blue in Figure 5 are the 2D models of cells created from SEM photos (Figure 5a–d). Finally, by placing these 2D models at their appropriate levels, a rough 3D model was created. Figure 5e shows an example for PA4. The preliminary 3D models were enhanced in accuracy by using foam density and cell volume as two control factors to compare with the experimental foam structure as follows: (a) A bank of spherical cells was created based on the cell size distribution (statistical population) derived from the statistical analyses of SEM data and histograms (Figure 1). (b) Spherical cells were randomly placed into the prototype 3D model from the data bank. (c) These cells are depicted in dark blue in Figure 5f. To validate the model, the foam density was applied as a control parameter. Inserting these complementary cells was performed until the model density was close to the target value (experimental data). The rubber matrices were subtracted from the real cells and adjacent spheres, resulting in pores in the 3D geometry. Figure 6a illustrates the 3D geometric models of the foams for different foaming agent concentrations (2, 3, 4 and 5 phr) based on 2D SEM images and foam density. The analysis of the POE foams was performed using the static structural system of the ANSYS software. In the meshing process (Figure 6b), a 3D model of each foam with its own meshing conditions was performed. The mesh type selected for these models is the “*mechanical solid quadratic tetrahedral*”, and the final meshed structure of the foams is presented in Figure 6b. Even with modern supercomputers, the computational load for the simulation of heterogeneous or complex structures, such as blends, composites and foams, can be very intensive, and calculations can take a long time while using a high amount of memory. 

In the theory of porous materials for hyperelastic material, the representative volume element (RVE) or representative element volume (REV) is the smallest unit of the whole that can be considered as a representative of the material for analysis and measurement. Since hyperelastic porous materials are not homogeneous materials, it is not possible to simply choose a periodic unit cell as a RVE, and the selection of these units is statistical and random. Therefore, for the analysis of porous hyperelastic materials in a continuous physical environment, a volume can be selected that statistically and effectively includes sampling of all the structural heterogeneities of these materials, including different cell sizes, cell orientations and cell densities [65,66].

In this work, after making POE foams and providing SEM images of their sections, a statistical and morphological analysis was performed on the cells in the structure of the foams. The desired RVE was selected based on these statistical results for the spectrum of cell geometric characteristics in the structure of the SEM images, in order to reflect the real structure of the foams as much as possible. This RVE was selected with the desired dimensions of 1 mm × 1 mm × 1 mm from the central part of the SEM images, and the cells obtained from the morphological analysis were used to approximate the 3D model from 2D information. Figure 6c depicts the final RVE of the 3D geometry model for PA4.

### 4.2. Boundary and Test Conditions

To represent the uniaxial tensile properties, the lower edge (bottom) of the foam was fixed, while the upper edge (top) moved in the vertical direction at a constant speed in a quasi-static mode. The foams’ lateral edges (sides) are regarded as symmetrical (periodic) boundary conditions. In this situation, the material’s influence on the location near to these edges is taken into account (which is not modeled). As a result, it is more realistic than using a free edge boundary condition [67].

### 4.3. Material Definition in FEM Models

As mentioned before, the finite element calculations were made using the ANSYS software. To analyze hyperelastic materials, in the first step, the mechanical behavior of the desired material must be defined in the software based on a specific pattern and the models of hyperelastic materials. 

The models of hyperelastic materials are equations based on the strain energy density. All the work carried out in deforming hyperelastic materials is conserved, and it is stored as internal energy, which is fully recovered upon unloading. This attribute of such materials is used in defining their constitutive model in terms of strain energy density, which is the amount of internal energy stored in a unit volume of the material. The strain energy density is a function of the strain in the material and the stresses developed in it are calculated based on the mode of deformation, so the response can differ, depending on how the material is deformed. 

To summarize, because of the complex stress–strain relationship of hyperelastic materials, hyperelasticity is typically defined with an equation, called strain energy potential. The main challenge is to determine the coefficients to characterize our material property. This is where experimental data from controlled experiments play a key role. Typically, we performed experimental tests (such as tensile test, compression test or shear test) and calibrate, or curve-fit, for the hyperelastic coefficients. The calibration of the material model involves selecting the hyperelastic model and curve-fitting it to the available experimental data. As a result, the constants are obtained, which are considered as material properties. 

In recent years, various hyperelastic models have been introduced and are available in the ANSYS software that a user can select from and perform curve fitting. These models includes Mooney–Rivlin, Blatz–Ko, polynomial and Yeoh, as well as Ogden 1st, 2nd and 3rd order [68,69]. By examining each model, researchers must select the most suitable model for their hyperelastic materials and measure the mechanical properties of these materials to obtain values for the parameters. In order to have an appropriate selection among the hyperelastic models for our POE foams, several hyperelastic models were considered for the neat matrix (PA0). Figure 7 presents the responses of each model under static loading to compare with the experimental results.

Here, as an example, the process of curve fitting and finding the material constant of the strain energy density equation is briefly presented for the Mooney–Rivlin 5 Parameter as a sample of hyperelastic models. From Equation (6), the strain energy equation in the five-parameter Mooney–Rivlin model can be written as:U = C_10_(I_1_ − 3) + C_01_(I_2_ − 3) + C_20_(I_1_ − 3)^2^ + C_11_(I_1_ − 3) (I_2_ − 3) + C_02_(I_2_ − 3)^2^(6)

As shown in Figure 8, based on the experimental results for the tensile test of the curve fitting, the constants of the material are obtained and presented in Table 3. Using these constants, the energy equation is completed and the deformation behavior of the hyperelastic material can be defined. 

As reported above, each of the hyperelastic materials must have its own mechanical properties and, as a result, it should be defined with its appropriate hyperelastic model. In this work, different models for the POE have been examined in order to select the best one for the finite element analysis. By calculating the errors (RMSE = Root Mean Squared Error and MSE = Mean Squared Error, as reported in Table 4), it can be concluded that the Mooney–Rivlin 5 Parameter (RMSE= 0.1023 and MSE = 0.0104) is the most suitable hyperelastic model for our POE. It can be seen in Figure 7 that the experimental data in the region between 0 and 0.05% of strain includes many degree changes. Therefore, the only hyperelastic model that could have the best fitting in this area was selected (Mooney–Rivlin 5 Parameter). In the rest of the points (0.05 to 0.2 strain) a good fit was obtained.

### 4.4. The Limitation of Problem Solving by FEM

Although there are many advantages and possibilities in the analysis of geometric problems with the help of a finite element, some limitations also exist. In this work, the problem-solving approach with the help of a finite element was divided into two main parts: (1) The generation of a geometric model. (2) The generation of a finite element model and its solution.

In the first part of the geometric model, two-dimensional SEM photos were used, and after analyzing the morphology with the help of the obtained results, the difficulty was in creating the three-dimensional model based on two-dimensional SEM images and to create a structure that was close to the real one. In this process, although factors such as the cell size and the cell shape help to make the statistical representation of the 3D model very close to reality, there is the possibility of other parameters being important, such as cell geometry and orientation. Today, with the advancement of X-ray technology, it is expected that more realistic 3D models of materials with all geometrical aspects of their substructures can be produced at a lower cost.

The second part includes the process of building the finite element model and solving it, although selecting boundary and loading conditions very close to reality has been attempted, but since a considered RVE is a part of the larger experimental model, it eventually causes errors in the problem-solving process. 

Another part of the finite element model is defining the material and assigning it to the finite element model. As mentioned in Section 4.2, there are errors related to the curve-fitting process, in some parts of which the problem-solving equation does not match the experimental results. It should be noted that after examining various hyperelastic models, every effort has been made to reduce this error to its lowest value.

### 4.5. The Results of FEM

FEM was used to investigate the stress–strain behavior of the foams under uniaxial tension at two levels: microscopic and macroscopic. Figure 9 illustrates the structural response under uniaxial tension loading compared to the unloaded structure for POE foams with different ADC contents. Based on these findings, the deformation level varies throughout the vertical position. The lowest deformation level without deformation (blue) occurred near the bottom plate, while the highest deformation level with the most distortion (red) occurred close to the moving top plate. Between both plates/limits, the deformation continuously varies, resulting in a deformation gradient across thickness due to the testing conditions imposed.

The macroscopic stress–strain behavior of the POE foams is presented in Figure 10. Two major pieces of information can be obtained at the macro-level:(a)The FEM results coincide well with the experimental results. After analyzing the experimental results, the average value of their RMSE is 1.5, and the percentage of errors between the experimental and numerical results is 9.2%. These results show the validity of the FEM results compared to the experimental results in the field of engineering standards. This can be attributed to a number of factors: fine material definition with hyperelastic models and an accurate 3D-modeled cell structure.(b)The FEM simulation and experimental results both showed that a relation between stress, strain and ADC content exists. For example, at 4 MPa stress, PA2 has the lowest strain (12%) and PA5 has the highest strain (15%). Similarly, PA2 requires a greater stress (4.9 MPa) than PA5 (3.8 MPa) to reach a strain of 15%. As mentioned before and according to these findings, the tensile properties of the foams are closely related to the foam density and less to the morphological parameters.

Apart from examining the stress–strain curves at the macroscopic level, one of the key advantages of FEM is that it can depict and evaluate the results at the microscopic level, which is difficult to study with hyperelastic models and impossible with direct experimental measurements. Figure 11 reports the maximum stress concentration in the foams for the microscopic level analysis. At the microscopic level, it can be seen that a POE foam with a higher foaming agent content (PA5) has a higher maximum stress concentration. The main reason is because of lower density, larger cell sizes and more cell connections (coalescence and collapse) generating a less homogeneous stress distribution, leading to more stress concentration around these structural defects. On the contrary, PA4 has a less maximum stress concentration owing to its lower cell size and less structural defects (cell connection). 

Figure 12 reports on the stress concentration in a small volume, which consists of closed cells and connected open cells. The stress concentration is the lowest (dark blue) while the cells are far away but gradually increases (color shifts from light blue to light green) as the cells move closer. Moreover, open or connected cells (yellow and green) create higher stress concentrations than closed or separated cells (light blue or light green). Furthermore, the stress distribution is mostly centered in the connected area between the distinct sections of two connected cells (red color). It is important to note that the amount of stress concentration is closely related to structural defects. As a result, components that are far away from the cells have less effect and display a low degree of stress concentration. It is important to note that the level of stress concentration is directly related to structural defects. However, upon moving closer to the cells, their influence grows stronger, causing more tension to build up. The structural defects are greatest at the cell connections (hot spots); therefore, the tension increases accordingly. Figure 13 shows that PA3 has the lowest micro-scale stress compared to other samples. According to the SEM images and morphological results (Figure 1), the dispersion of cells in PA3 is better compared to PA2. In other words, the cells in PA2 are more evenly distributed all over the matrix. Furthermore, the cells in PA3 were arranged more regularly, resulting in a reduced stress flow and less stress concentrations in the microstructure. Increasing the ADC content from 3 to 5 phr leads to higher levels of microscale stress (for the same microscale strain). For example, the stress concentrations at 30% strain for PA2, PA3, PA4 and PA5 based on the FEM results are 16.8 MPa, 14.1 MPa, 20.1 MPa and 23.2 MPa, respectively. However, foams containing more foaming agents have larger cells, lower density and more cell connections (coalescence and collapse), resulting in less homogenous stress distribution and more stress concentration around structural defects. As a result, a higher foaming agent content leads to weaker substructures that are more vulnerable to different pressures, higher failure modes and more internal substructure deformation due to higher stress and strain concentration.

## 5. Conclusions

This work investigated the effect of azodicarbonamide (ADC) content (2–5 phr) on the structure and properties of a polyolefin elastomer (POE) foam. The samples were produced by a single-step compression molding process, leading to foam density in the range of 0.61–0.75 g/cm^3^ compared to the unfoamed matrix (0.87 g/cm^3^). The quantitative analysis of the SEM images showed that the sample with 4 phr of ADC had the best morphological properties with the lowest cell size (109 μm), highest cell density (590 cells/mm^3^) and narrowest cell size distribution (PDI = 1.037). This optimum structure (ADC content) represents a balance between a high amount of gas generated (high cell nucleation and cell growth), while limiting cell collapse, coalescence and rupture. The tensile and hardness properties followed the density, as they all decreased with an increasing ADC content. On the other hand, increasing the ADC content improved the compression behavior, as more gas generated a higher internal cell pressure acting against the imposed compression stresses. A further analysis of the results showed that the tensile and compression properties are not directly related to the same parameters. In fact, the tensile properties are more related to foam density (foaming ratio), while the compressive properties are more dependent on the cellular structure (cell size, cell density and internal gas pressure). Finally, the 3D foam structures obtained by a combination of 2D SEM images were used to predict the mechanical properties of the resulting foams. The FEM analysis at two levels (macro- and micro-scale) was performed, and the results were in good agreement with the experimental results for the range of conditions tested. While the micro-scale results showed that the lowest stress levels were obtained for the sample with 3 phr of ADC, the highest micro-scale stress levels were observed for the sample with 5 phr of ADC, due to their lowest and highest cell sizes, respectively. The different trend of PA3 is due to the low cell size and better cell dispersion inside the matrix. The FEM analysis of other mechanical properties was not addressed in this work to limit the scope and the paper length. Nevertheless, they are currently under investigation and the results will be reported in a future publication. Finally, other numerical methods, such as the differential quadrature method, will be used to model the heat transfer behavior of POE foams as thermal insulators.

## Figures and Tables

**Figure 1 polymers-14-04692-f001:**
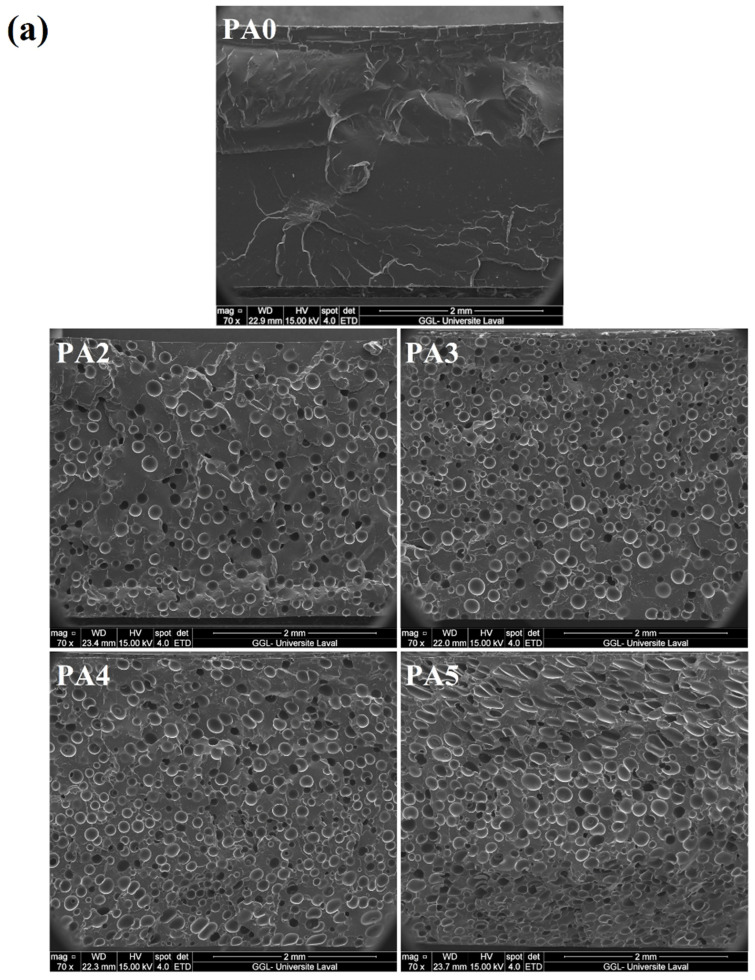
(**a**) SEM images of the POE foams with different ADC contents; (**b**) effect of ADC content on the cell size distribution of POE foams.

**Figure 2 polymers-14-04692-f002:**
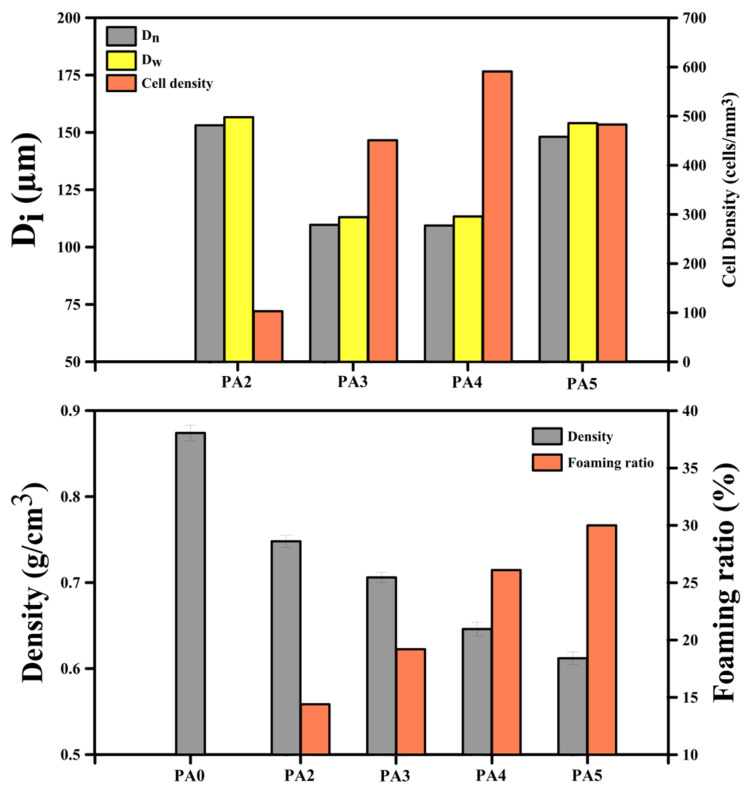
Morphological and physical characteristics of POE foams with different ADC contents.

**Figure 3 polymers-14-04692-f003:**
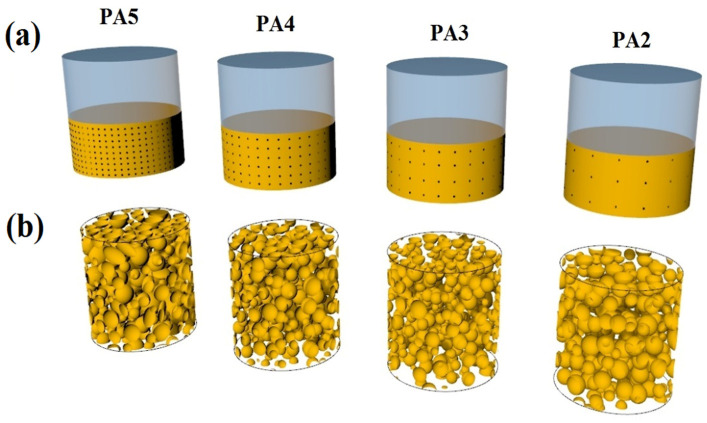
Schematic representation of the foaming process with different foaming agent contents’ ((**a**) first row) nucleation with concentration and (**b**) final structure.

**Figure 4 polymers-14-04692-f004:**
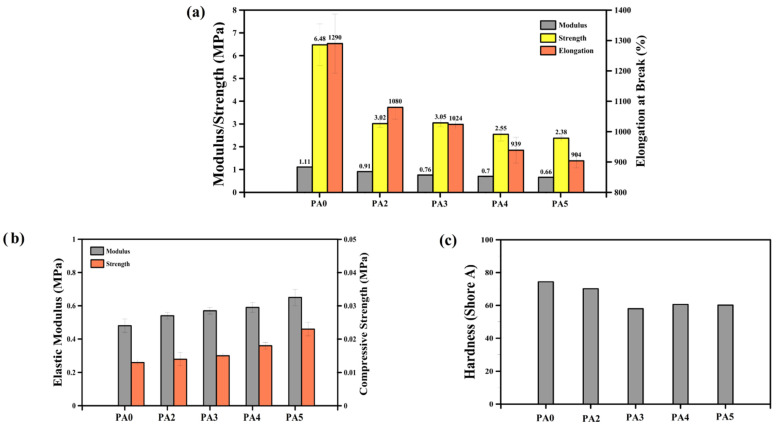
(**a**) Tensile, (**b**) compression and (**c**) hardness of POE foams with different ADC contents.

**Figure 5 polymers-14-04692-f005:**
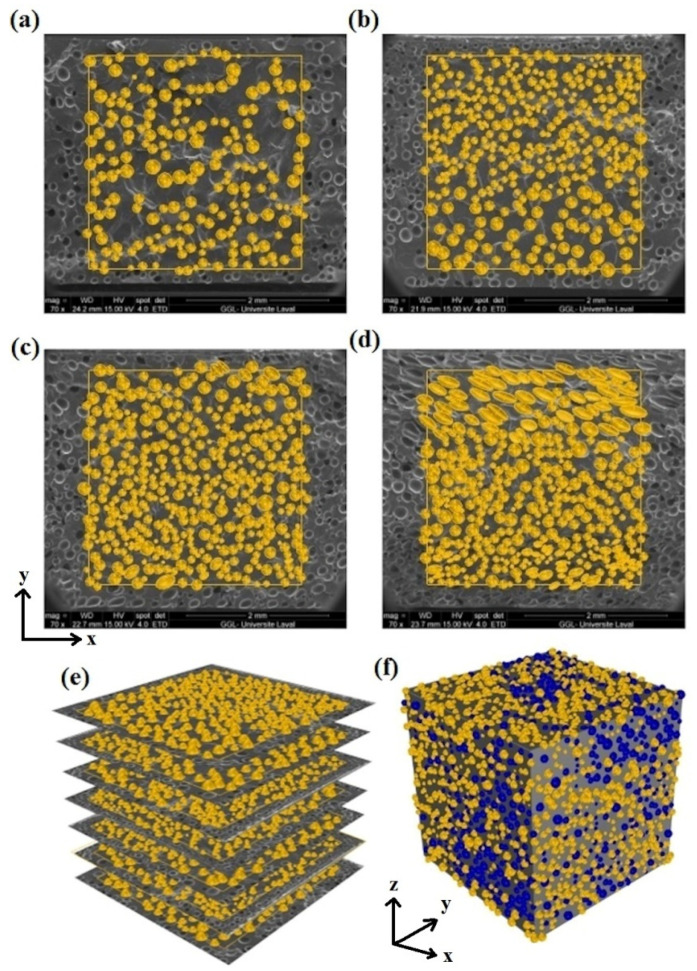
Real 2D geometry models from SEM images for POE foams: (**a**) PA2, (**b**) PA3, (**c**) PA4, (**d**) PA5, as well as (**e**) the real 3D geometry based on 2D images for PA4 and (**f**) the final 3D structure for PA4.

**Figure 6 polymers-14-04692-f006:**
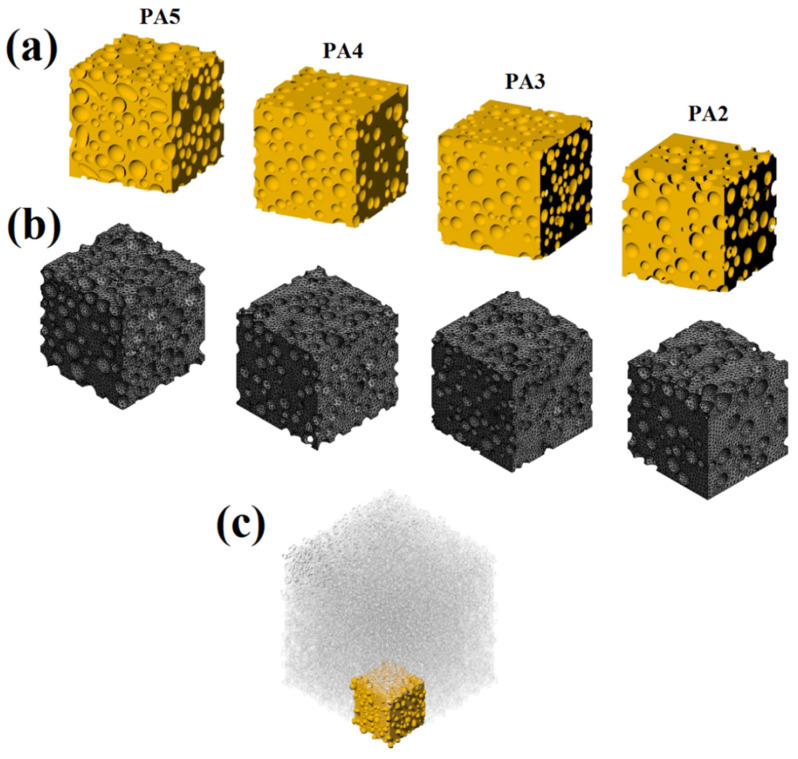
(**a**) Final 3D geometric models, (**b**) meshed geometric model and (**c**) the RVE used.

**Figure 7 polymers-14-04692-f007:**
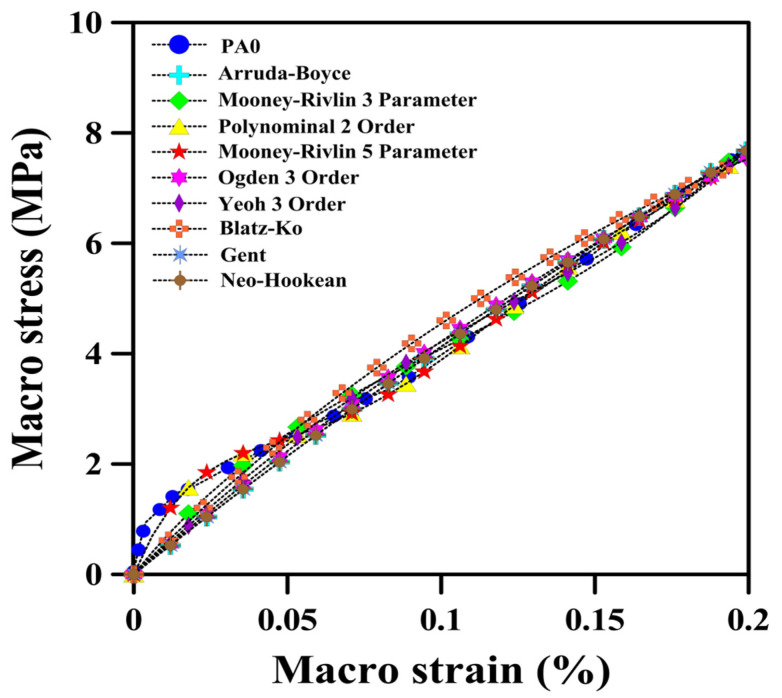
Mechanical behavior of the POE with different hyperelastic models under static loading.

**Figure 8 polymers-14-04692-f008:**
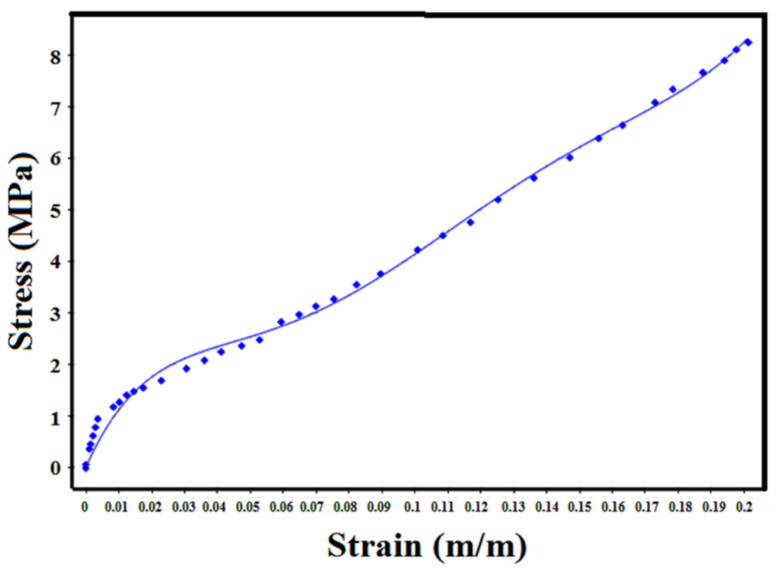
Curve fitting based on experimental tensile test.

**Figure 9 polymers-14-04692-f009:**
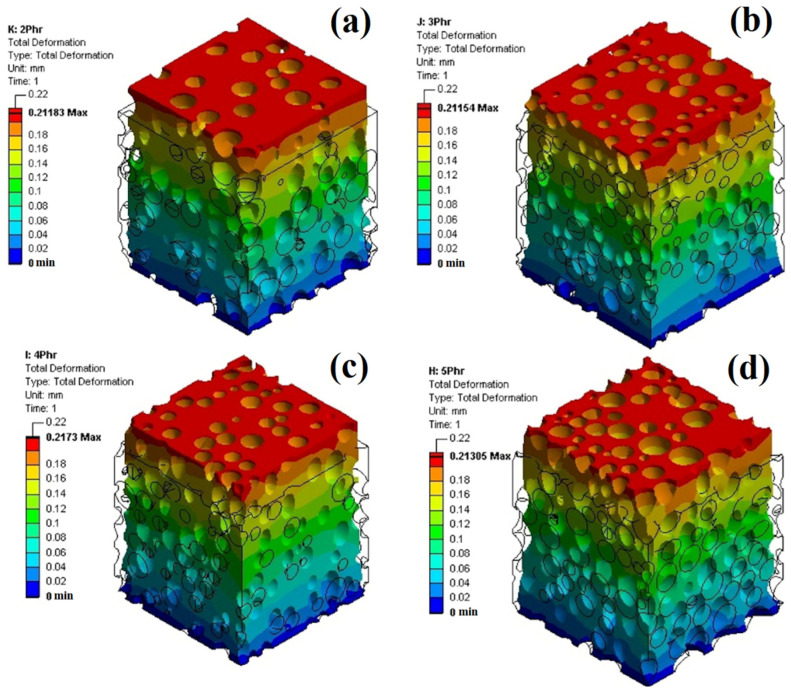
Structural response of POE foams under uniaxial tension loading, after loading (colored structure) and before loading (black line) for foams with different ADC contents: (**a**) 2 phr, (**b**) 3 phr, (**c**) 4 phr and (**d**) 5 phr.

**Figure 10 polymers-14-04692-f010:**
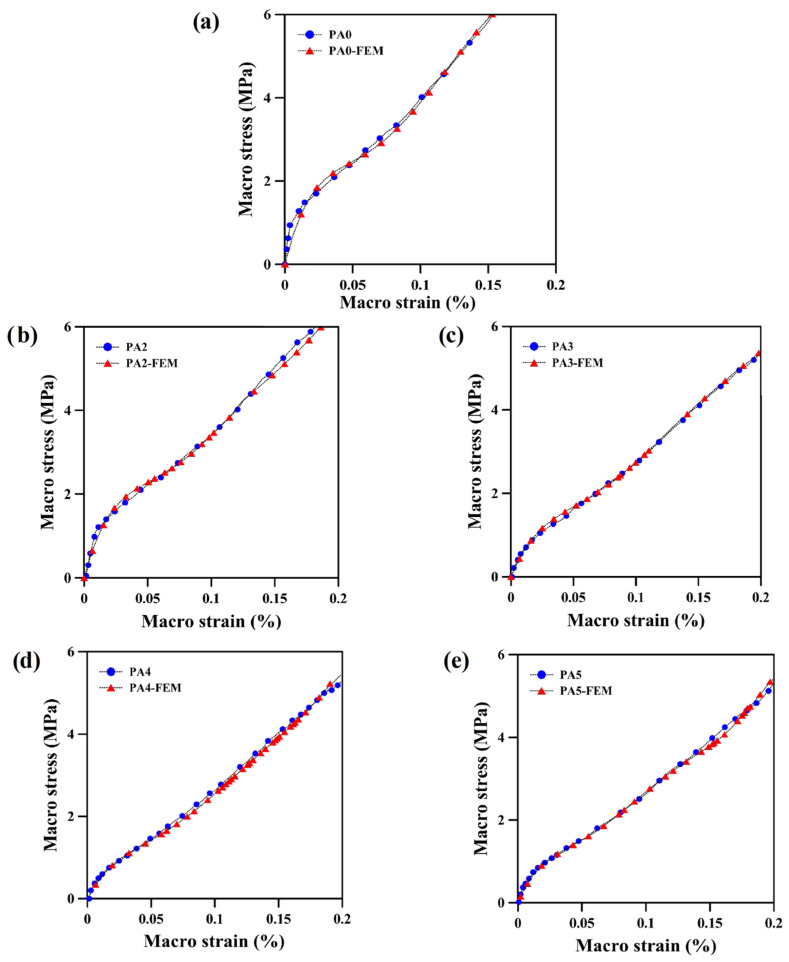
Macroscopic stress–strain behavior of: (**a**) unfoamed and foamed POE with different ADC contents: (**b**) 2 phr, (**c**) 3 phr, (**d**) 4 phr and (**e**) 5 phr.

**Figure 11 polymers-14-04692-f011:**
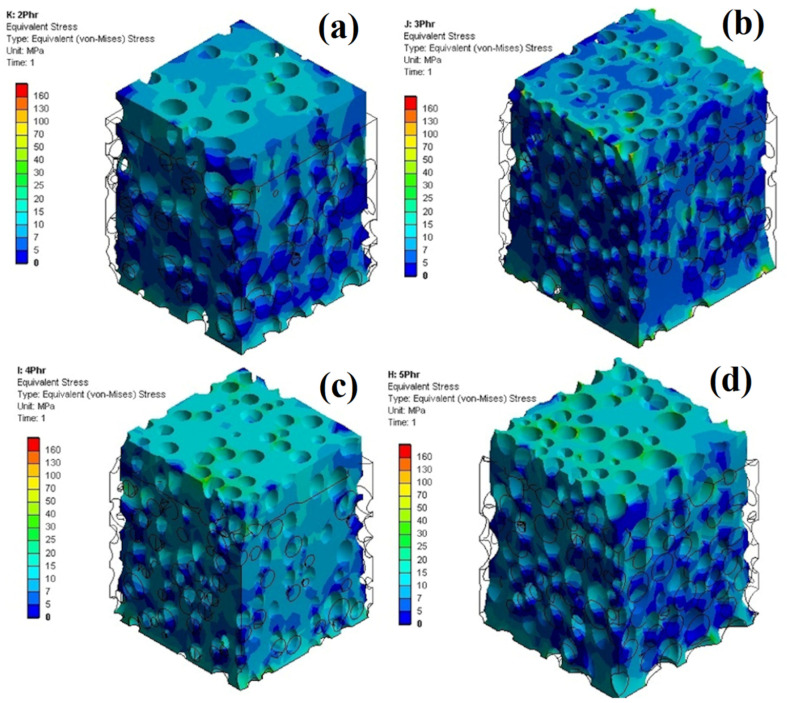
Maximum stress during uniaxial tension for POE foams with different ADC contents: (**a**) 2 phr, (**b**) 3 phr, (**c**) 4 phr and (**d**) 5 phr.

**Figure 12 polymers-14-04692-f012:**
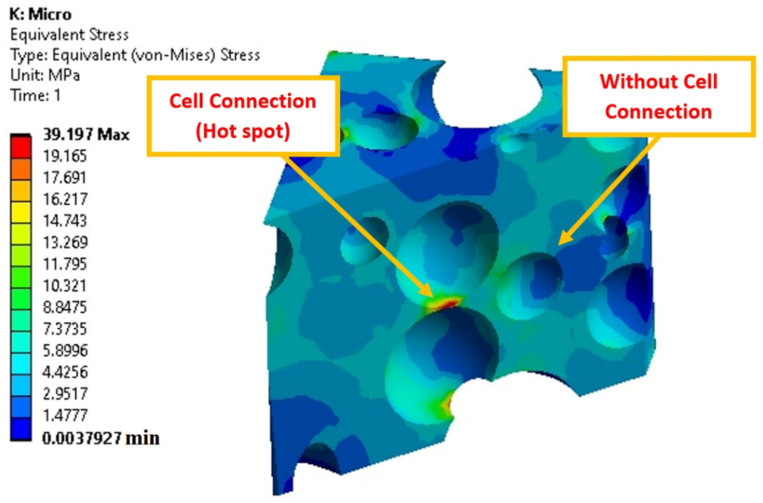
Stress contour at structural defects (PA4).

**Figure 13 polymers-14-04692-f013:**
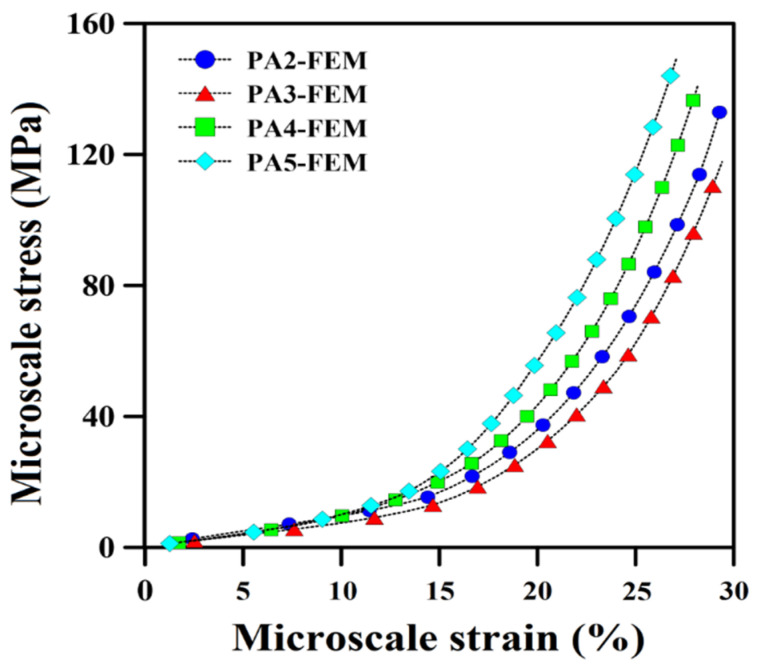
Micro-scale stress as a function of micro-scale strain for the POE foams.

**Table 1 polymers-14-04692-t001:** The samples’ codes used with their composition.

Sample Code	PEO (phr)	ADC (phr)
PA0	100	0
PA2	100	2
PA3	100	3
PA4	100	4
PA5	100	5

**Table 2 polymers-14-04692-t002:** The equations and features of the morphological characteristics [56,57,58].

	Main Equation	Feature	Equation
Number average cell size	$D_{n}=\frac{\sum^{\;}\left(n_{i}\;D_{i}\right)}{\sum^{\;}n_{i}}$	Number average diameter of cells	(2)
Weight average cell size	$D_{w}=\frac{\sum^{\;}\left(n_{i}\;D_{i}^{2}\right)}{\sum^{\;}(n_{i}\;D_{i})}$	Weight average diameter of cells	(3)
Polydispersity index	$PDI=\frac{D_{w}}{D_{n}}$	Cell size distribution	(4)
Cell density	$\rho_{cell}=\frac{\rho_{bulk}}{\rho_{foam}}\;{\left(\frac{\sum^{\;}n_{i}}{A}\right)}^{\frac{3}{2}}$	Population and number of cells inside the matrix	(5)

*n_i_* is the number of cells with a diameter *D_i_*, *A* is the surface analyzed of the foam, *ρ_bulk_* is the density of the unfoamed matrix and *ρ_foam_* is the foam density. For hardness (Shore A), a PTC Instrument (ASTM D2240, model 307 L, PTC Instruments, Los Angeles, California, USA) was used. Tensile testing (ASTM D412) was performed at room temperature on an Instron universal testing machine (USA) model 5565 with a 500 N load cell at a crosshead speed of 10 in/min. For the compression tests (ASTM D575), a dynamic mechanical analyzer (DMA) RSA3 TA Instrument (New Castle, DE, USA) was used with cylindrical samples (2.5 cm in diameter and 3.5 mm thick) and compressed at a rate of 0.01 mm/s. All the properties reported represent an average of a minimum of three samples.

**Table 3 polymers-14-04692-t003:** Material constants for Mooney–Rivlin 5 Parameter (Equation (6)).

Material Constant	Value
C01	489.3824
C02	13,268.73
C10	−467.077
C11	−20,653.2
C20	8267.356

**Table 4 polymers-14-04692-t004:** RMSE and MSE for the different hyperelastic models.

Hyperelastic Model	Mean Square Error (MSE)	Root Mean Square Error (RMSE)
Arruda–Boyce	0.1029	0.3208
Mooney–Rivlin 3 Parameter	0.0542	0.2329
Polynomial 2nd Order	0.0104	0.1024
Mooney–Rivlin 5 Parameter	0.0104	0.1023
Ogden 3rd Order	0.1011	0.3179
Yeoh 3rd Order	0.0836	0.2892
Blatz–Ko	0.1257	0.3545
Gent	0.1031	0.3211
Neo-Hookean	0.10294	0.32083

## Data Availability

The data presented in this study are available on request from the corresponding author.

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
