# Peer review of "Experimental and Finite Element Simulation of Polyolefin Elastomer Foams Using Real 3D Structures: Effect of Foaming Agent Content"

_polymers, 2022, doi:10.3390/polym14214692_

Round 1

Reviewer 1 Report

1. Equation 1 is a definition that is usually used as a foaming ratio or porosity or void fraction.The volume expansion ratio is the ratio of the volume of the foamed specimen to the volume of the foamed specimen.

2. In the experiment, tensile, hardness, and compression were conducted, but why only tensile was conducted by FEM analysis?Changes in tension, hardness, and compression according to the contents of ADC are like the results of many studies so far and cannot be seen as new contributions.

Therefore, it is necessary to conduct FEM analysis using not only tension but also compression and hardness using the model to verify that the model can be used in all analysis processes.

3. In terms of PDI verification, it is advantageous to proceed with non-dimensionalize the y-axis of Figure 1.a) with the total number of cells in the experiment

4. The originality of this paper is real structure modeling, so a clear explanation is needed.

According to the scale of the SEM image in Figure 3, it is determined that the model has an image dimension of about 4 mm x 4 mm.

The FEM analysis of this experiment was conducted using a representative element volume(REV), but there is no explanation for how rev is appropriately selected.

This process is a very critical part of the paper, so it is necessary to write in detail what the factor for selecting an appropriate REV is and how it was optimized.

5. In the case of Figure 3.d (PA5), it is confirmed that there is a gradation of the cell size even with a position difference of 3 mm.

There seems to be a tendency of cell morphology depending on the location, and if the entire model is made without considering the location of the layer,

such as Figure 3e) using REV, it seems that there is no difference from the model that randomly made without reflecting the morphology.

6. The basic description of ? or I, which is mainly used in the hyperelastic model, should be included in the paper.

If the Nomenclature part is not to be inserted, an explanation must be added to the table side.

7. The legend in Figure 6 needs to be modified. Write exactly where it doesn't overlap.

8. Please unify all the legends in Figure 7 and make them check the difference in deformation according to the ADC content through color.

9. There is a need to unify the legend in Figure 9. As in the previous comment, it should be possible to check the tendency according to each condition by color.

10. When checking Figure 9, it seems that dark blue color is the most in PA5.

Therefore, the explanation of Line 389 393 does not fit with Figure 9.  It is necessary to show the maximum stress concentration numerically.

11. In Figure 11, it is necessary to explain why PA3 phr has different tendencies.

12. According to the explanation of this paper, Microscale is a problem with cell location rather than expansion ratio.

Then, isn't this experiment not a general result based on the foaming agent content, but a typical result of the specimen model?

typo

1. In line 134, 140 and 141, PEO should change to POE

2. Remove typo in equation 7 and 10 in Table 4.

Reviewer 2 Report

Please read and “fully” address the comments listed below:

  1. The ABSTRACT is not written in a logical order. Start with an overview of the topic and a rationale for your paper. Describe the methodology you used and the general outline of the manuscript. Also, in the end, state the result in more detail (i.e., provide some numbers).

  2. The novelty of your work is still unclear to the reader, which should be detailed both in the Abstract and Introduction.

  3. In Figs. 7, and 9, please change the color map from rainbow/ jet to a perceptually uniform (color-blind friendly) scale.

  4. Please add an error bar to all your bar charts, where applicable.

  5. Please fully introduce the elastic properties of all the structural components explained in section 2. You can summarize them in a table.

  6. Include XYZ coordinates to all figures, where needed, e.g., in Figs. 1, 2, and 3.

  7. Please fully explain the process used for mixing/ casting the specimens.

  8. Please provide more explanation for this sentence: Page 16, Line 416: "But foams containing more foaming agents have larger cells and more cell connections, resulting in more structural defects and stress concentration."

  9. Did you do any sensitivity analysis to determine the optimum mesh size for your FEM analyses? Can you provide the subroutine of the FEM code in the appendix of your paper (in 10-15 lines) such that future readers can replicate your work?

  10. Apart from FEM analysis, more novel and strong numerical methods have been recently proposed to find the stress-strain properties of composites. Among them, the “Differential Quadrature” and “Bezier” methods proved to have higher stability and accuracy than other numerical methods. For this purpose, please write a paragraph in your paper introducing these methods which can “alternatively” provide the stress analysis of your composites: 

    Differential Quadrature Method: 

  • Yan, Y., Liu, B., Xing, Y., Carrera, E., & Pagani, A. (2021). Free vibration analysis of variable stiffness composite laminated beams and plates by novel hierarchical differential quadrature finite elements. Composite Structures, 274, 114364. 

    Bezier Method:

  • Kabir, H., & Aghdam, M. M. (2021). A generalized 2D Bézier-based solution for stress analysis of notched epoxy resin plates reinforced with graphene nanoplatelets. Thin-Walled Structures, 169, 108484.

11. Conclusion: Can authors highlight future research directions and recommendations? Also, highlight the assumptions and limitations (e.g 1-2 shortcoming(s) of the present study). Besides, recheck your manuscript and polish it for grammatical mistakes (you can use “Grammarly” or similar software to quickly edit your document).

Author Response

Please see the attachment."

Reviewer 3 Report

Thank you for submitting your paper. The work done here draws attention to a significant subject modelling of elastomer foams performance and experimental validation. I have found the paper to be interesting. However, several issues need to be addressed properly before the paper is being considered for publication. My comments including major and minor concerns are given below:

Please consider reviewing the abstract and highlight the novelty, major findings, and conclusions. I suggest reorganizing the abstract, highlighting the novelties introduced. The abstract should contain answers to the following questions:

What problem was studied and why is it important?

What methods were used?

What conclusions can be drawn from the results? (Please provide specific results and not generic ones).

The abstract must be improved. It should be expanded. Please use numbers or % terms to clearly shows us the results in your experimental work.

Please consider reporting on studies related to your work from mdpi journals.

The authors should remove all bulk citations, unless given full credit afterwards. The authors should check for this issue elsewhere in the manuscript.

Check symbol degree format such as in line 153

How many samples in total were produced? How many repetitions? Is three samples enough!

Authors should add images of the experimental setup and produced/used samples for testing/analysis. After all this is part experimental analysis and graphical details are needed to give the readers better idea of what was done in the experimental part.

4. Modeling and simulation this section should be after section 2 Materials and methods then the results and discussion section should follow.

In lines such as line 184 and 195 authors need to replace [ ] with ( ) to avoid confusion with referencing numbers. Authors should check this issue elsewhere in the manuscript.

In page 5, the large paragraph is basically increase and decrease observations of the results with no critical discussion or proper comparison with literature.

Authors need to bring figure 1 before Table 2 as it was mentioned first.

Table 2 should be removed and replaced with bar chart graph. It’s much easier to interpret data from bar chart graphs for comparison.

Lines 209-210 “This trend is related to the higher amount of gas generated with increasing ADC content.” Is this a claim or a fact? In either way authors need to support this claim with a reference(s)

Line 198 “higher plasticization effect” did the authors actually see that and compared it to different samples? Authors are making a lot of claims but it is not clear if this was observed or just speculations?

Figure 1 authors need to add some text and arrows to clearly show us what to look at in those SEM images.

Line 238 “the compressive stress-strain curves have a different trend with increasing foaming agent content” can the authors support this claim with a reference. What does the literature similar to your work say? Are they getting similar results or different trends from yours?

Table 3 again authors should use bar chart graphs and add error bars for better interpretation of the results.

Authors need to mention limitations in their FE model compared to previous ones. Boundary conditions and the source(s) of material properties and other parameters used in the model.

Authors need to remove Table 4 as it is not suitable to add in a research paper, it is more suited in a thesis or a review paper instead. Or perhaps move to an appendix?

Table 5 if data not produced by authors, then it should be referenced.

4.4. Finite Element Analysis (FEM) authors need to improve the title of this subsection

Line 372 please use % or numbers to clearly to tell the readers how well the FE and Exp results agree with each other. a bar chart or graph would be helpful.

What could be reason behind differences in results accuracy between FE and Exp? mention how the model can be improved. 

 Conclusion can be expanded or perhaps consider using bullet points (1-2 bullet points) from each of the subsections.

Round 2

Reviewer 2 Report

The authors addressed my comments and the manuscript can be published in the present format.

Reviewer 3 Report

The authors have answered all questions and paper can be accepted.